# Metastasis in Pancreatic Ductal Adenocarcinoma: Current Standing and Methodologies

**DOI:** 10.3390/genes11010006

**Published:** 2019-12-19

**Authors:** Marina Ayres Pereira, Iok In Christine Chio

**Affiliations:** Institute for Cancer Genetics, Department of Genetics and Development, Columbia University Medical Center, New York, NY 10032, USA; mma2246@cumc.columbia.edu

**Keywords:** pancreatic cancer, metastasis, metastasis models

## Abstract

Pancreatic ductal adenocarcinoma is an extremely aggressive disease with a high metastatic potential. Most patients are diagnosed with metastatic disease, at which the five-year survival rate is only 3%. A better understanding of the mechanisms that drive metastasis is imperative for the development of better therapeutic interventions. Here, we take the reader through our current knowledge of the parameters that support metastatic progression in pancreatic ductal adenocarcinoma, and the experimental models that are at our disposal to study this process. We also describe the advantages and limitations of these models to study the different aspects of metastatic dissemination.

## 1. A Brief Introduction to Pancreatic Cancer

### 1.1. Epidemiology and Clinical Outcome of PDA

Pancreatic cancer can either be exocrine or neuroendocrine (endocrine tumors), depending on the cell of origin. About 93% of pancreatic cancers are exocrine tumors, the most common type being pancreatic ductal adenocarcinoma (PDA). The remaining 7% are neuroendocrine tumors (PNET), also called islet tumors, which often grow more slowly than their exocrine counterparts [1,2]. In addition to the low frequency of cases, PNET are mostly characterized as indolent. While 40%–80% of patients with PNET are metastatic at presentation, usually involving the liver (40%–93%) [3], treatment options do exist and include locoregional therapy, chemotherapy, as well as liver transplant [4]. A detailed review on the metastasis of PNET can be found in [5]. Herein, we will focus on the metastasis of PDA.

PDA is a highly aggressive malignancy with limited treatment options and a dismal prognosis. PDA represents the fourth leading cause of cancer death worldwide, and the 12th most common cancer in the world [6,7]. The detrimental outcome is related to delayed diagnosis, often a consequence of non-specific symptoms such as abdominal pain, jaundice and weight loss. The retroperitoneal location of the pancreas also means there are no external lumps that can be palpated during an annual routine physical exam, such as may be the case for breast cancer. Further, the aggressive nature of the disease is associated with a high potency of metastatic dissemination to adjacent organs such as the liver and the gallbladder [7,8,9], which is often already detected at diagnosis [10]. In these cases, surgery is rarely a viable option. Even with intended curative surgical resection of the primary tumor with no tumor margin (R0) and no evidence of metastasis at resection, 75% of the patients die of metastatic disease within 5 years after surgery [7,11]. 

The current standard of care for patients with PDA includes chemotherapeutic cocktails that are highly toxic with limited specificity. Despite many attempts to optimize the chemotherapeutic regimens for PDA in clinical studies, the increase in the overall survival rate is poor. As the majority of PDA patients die of metastatic disease, this underscores the urgent need to develop novel therapeutics that targets not just the primary tumor but also the biological vulnerabilities of metastatic PDA cells.

### 1.2. Genetic and Molecular Classification of PDA

Oncogenic activation of *KRAS* is the most frequent genetic alteration in PDA (>90%) [12]. While activating mutations of *KRAS* downstream signaling pathways, including BRAF-MAPK and PI3K-AKT, have also been observed, they are less frequent [13,14,15]. Mutations of tumor suppressor genes found in PDA include *CDKN2A/p16* [16], *TP53* [17,18], and *SMAD4/DPC4* [19,20]. Over 90% of early PanIN-1 have *KRAS* mutations, and mutations in *KRAS*, *BRAF*, *p16/CDKN2A* or *GNAS* are present in over 99% of early lesions [21]. Despite extensive genomic characterization, individual DNA mutations are yet to provide theranostic information for PDA. This has prompted efforts to perform in-depth molecular profiling of PDA to identify its transcriptional classifiers [22]. 

Using bulk tumor samples, several studies have identified various subtypes of ductal pancreatic tumor [23,24,25]. In general, it was found that PDA includes at least two groups distinguished by markers of epithelial differentiation state, with the more poorly differentiated (“basal-like”, “squamous”, or “quasi-mesenchymal”) exhibiting reduced survival relative to well-differentiated subtypes (“classical” or “progenitor”) [23,24,25]. More recently, these sub-classifications were unified by a study led by Maurer et al. in which laser capture microdissection RNA sequencing on PDA epithelia and adjacent stroma was performed [26]. This work revealed the presence of two tumor epithelial subtypes (basal and classical) and two activated stromal subtypes (immune signaling and matricellular fibrosis). Importantly, these results indicate the linkage between epithelial and stromal subtypes, thus revealing the potential interdependence of the evolution of tissue compartments in PDA [26]. This highlights the importance of understanding the biology of both the cancer cells and their surrounding microenvironment in the process of tumor progression and metastasis to advance therapeutic development and prognostication in the coming years.

## 2. Factors Governing Metastasis 

Next-generation genome sequencing of treatment-naïve pancreatic primary tumors and patient-matched metastasis has revealed that cells initiating distant metastasis are genetically identical, and that the different metastatic lesions share identical driver gene mutations [27]. This suggests that transcriptional or post-transcriptional changes are central to supporting the complex series of biological hurdles that must be surpassed for pancreatic cancer to metastasize [28,29]. These hurdles include detachment of the cancer cell from the basement membrane, invasion of surrounding tissue, intravasation (i.e., entering circulation), survival in circulation, extravasation into the parenchyma of distant tissues, and outgrowth into macrometastatic lesions. In PDA, it has been shown that metastasis can occur through early dissemination, even before the formation of a primary tumor mass [30,31]. Early disseminated cancer cells remain dormant with an increased resistance to current therapies [30,31] and exhibit clonal diversity on the basis of the site of metastatic invasion [32]. Specifically, lineage tracing analysis revealed that metastases in the peritoneum and diaphragm exhibit polyclonality, whereas those in the lung and liver tend to be monoclonal [32]. These observations suggest that heterotypic interactions between tumor subclones as well as site-specific selective pressures are both central to influencing metastatic initiation and progression. 

Dissemination of neoplastic cells can occur through the blood vessels or the lymphatic system. The latter usually involves the invasion of lymph nodes, starting with the sentinel node (i.e., the closest) [33]. Several factors determine the method of dissemination, including physical restrictions and accessibility of the different vasculature [33]. Here, we will focus on our understanding of metastatic events through the vasculature and summarize the important advances that have contributed to the identification of the factors involved in the dissemination and metastasis formation in PDA. 

### 2.1. Epithelial to Mesenchymal Transition and Invasion

In order for cancer cells to leave the primary tumor site and disseminate, they must acquire “pro-metastatic traits”. One of the most extensively studied “pro-metastatic traits” is the epithelial-to-mesenchymal transition (EMT), the transition of epithelial cells into motile mesenchymal cells, which plays an important role in embryogenesis, cancer invasion, and metastasis [34]. This process is associated with the loss of epithelial characteristics, including polarity and specialized cell–cell contacts, and the gain of a mesenchymal migratory behavior, allowing them to move away from their epithelial cell community and to integrate into surrounding or distant tissues [29,35]. In PDA, the EMT program has also been shown to increase tumor-initiating capabilities [36] and drug resistance [37,38,39]. More recently, it has been shown that the PDA EMT program consists of an intermediate cell state coined “partial EMT” [40,41,42,43]. The partial EMT phenotype is characterized by the maintenance of an epithelial program at the protein level, in contrast to a complete EMT phenotype which is characterized by the lack of epithelial marker expression both at the mRNA and protein levels [43]. Moreover, the partial EMT phenotype is characterized by the re-localization of epithelial proteins (including E-cadherin) to recycling endosomes. Interestingly, partial EMT cells migrate as both single and collective cells, in contrast to complete EMT cells that mainly migrate in isolation [43]. This is in contrast to the conventional notion that cells from connective tissue tumors such as fibrosarcoma and glioma tend to migrate individually, whereas cells from melanoma and carcinoma often migrate collectively [29,44,45,46]. The different modes of cancer dissemination (single *vs.* clusters) seem to influence the metastatic potential of cancer cells, as several studies have shown that tumor clusters have a higher metastatic potential than single cells [45,47,48,49]. Cell clusters can also be heterogeneous [50] and composed of cells from the tumor stroma co-migrating with cancer cells to distant sites. For example, pancreatic stellate cells (PSCs) co-injected orthotopically with pancreatic cancer cells can be identified in distant metastasis [51]. 

The induction of EMT by TGF-β was first recognized in cell culture [52]. The TGF-β-induced activation of the receptor complex leads to activation of SMAD2 and SMAD3 [53]. Activated SMAD2 and SMAD3 form a heterotrimer with SMAD4, and translocate into the nucleus, where they associate and cooperate with DNA-binding transcription factors to activate or repress the transcription of target genes such as *SLUG*, *SNAIL1* and *TWIST* [53]. Interestingly, in a genetically engineered mouse model (GEMM) of PDA, it was previously reported that SNAIL and TWIST may actually be dispensable for PDA dissemination and metastasis [54]. Other factors contributing to EMT include alterations in mucin expression [55]. Indeed, matched sets of tissues obtained from autopsy patients revealed significant differences in the expression of both membrane-associated and secreted mucins harboring different glycosylation modifications from early lesions to metastasis [56]. Sonic hedgehog (SHH) produced by the pancreatic epithelia, has been shown to enhance angiogenesis in vivo and contribute to tumor metastasis, particularly to the lymph nodes [57]. More recently, transcriptome and enhancer landscape profiling of murine primary tumors and metastatic organoids have suggested that enhancer reprogramming promotes pancreatic cancer metastasis and implicates *FOXA1* to be a major driver of invasiveness [58]. 

### 2.2. Surviving Oxidative Changes in the Circulation and Distant Parenchyma

Successful metastatic outgrowth requires cancer cells to undergo adaptations to survive the highly oxidative environment in circulation and in the parenchyma. The oxygen levels in peripheral tissues is reported to be around 38 mmHg (ranging between 57 mmHg and 30 mmHg depending on the tissue), whereas in arterial blood the concentration can be as high as 70 mmHg [59]. The oxygen tension in PDA is reportedly 19.1 times lower than that in normal tissue [60]. Therefore, disseminated PDA cells must engage adaptive mechanisms to survive this drastic change. 

Reactive oxygen species (ROS) are byproducts of cellular metabolism that could act as signaling molecules to regulate cellular metabolism itself [61]. There is a very delicate balance between the levels of ROS and their function as tumor promoters versus cell toxicity. While low levels of ROS can promote tumorigenesis [62], at high levels they can also induce cell death [63,64]. Extensive reviews on the role of ROS in cancer can be read elsewhere [65,66,67]. Previous studies demonstrate a role for ROS scavenging in protecting cells against anoikis [68]. Consistently, in experimental models of melanoma [69,70], breast [71], and lung cancer [72], treatment with antioxidants enhances metastasis. 

Within the primary tumor, it has been shown that PDA cells can engage different mechanisms to regulate the levels of intracellular ROS. These include increased glucose flux through the pentose phosphate pathway (PPP) [73,74] and increased glutamine metabolism [75,76]. Both of these mechanisms result in a net increase in the levels of the reducing equivalent NADPH, which maintains the levels of intracellular reduced glutathione [77]. PDA cells can also counteract the high levels of ROS accumulation by upregulating NFE2l2/NRF2, a master regulator of redox homeostasis. NRF2 has been shown to have a role in PDA initiation [78], tumor maintenance [79] and possibly metastasis [80]. The specific redox defense mechanisms involved in supporting PDA metastatic dissemination remains to be determined.

### 2.3. Interactions with the Tumor Microenvironment

The pancreatic tumor microenvironment (TME) is composed of stromal cells and extracellular matrix (ECM) components [81]. The predominant populations of stromal cells in PDA include cancer-associated fibroblasts (or activated PSCs), regulatory T cells, and tumor-associated macrophages [81]. Soluble factors from activated PSCs can prime the primary tumor for metastasis and cell migration. For example, PSC-derived hepatocyte growth factor (HGF) [82], insulin growth factor 1 (IGF-1) [82], and interleukin-6 (IL-6) [83] can induce EMT in PDA cells. In addition to facilitating EMT, secreted factors from PSCs such as matrix metalloproteases [84], collagen I [85], IL-6 [86], and galectin-1 [87] can also directly stimulate PDA cell migration. Besides directly activating pro-metastatic properties in the primary tumor, stromal factors also contribute to preparing a pre-metastatic niche in distant organ sites to facilitate the seeding of metastatic cells [88,89]. In an orthotopic model of PDA, it was shown that monocytes are recruited to the liver to prepare a supportive niche during cancer progression [90]. Similarly, hepatocyte-derived IL-1 contributes to creating an inflammatory environment to support PDA metastatic seeding and development in the liver [91]. The desmoplastic nature of PDA also results in limited oxygen availability [92]. The hypoxic environment that ensues has been shown to induce the expression of the transcription factor BLIMP in a subset of cancer cells, contributing to the tumor heterogeneity, and providing these cells with a transient metastatic potential [93]. 

Cancer stem cells (CSCs) characterized by the expression of CD133 and CXCR4 markers have been identified at the invasive front of pancreatic tumors [94]. These cells are characterized by a dependency on oxidative metabolism and reduced metabolic plasticity, determined by a decrease in c-MYC expression compared to non-CSCs [95]. Interestingly, this dependency on oxidative phosphorylation (OXPHOS) seems to be shared by a subpopulation of dormant tumor cells [96], thus raising the possibility of targeting these cells with OXPHOS inhibitors. As the availability of nutrients is different in each organ site [97], the metabolic adaptations required for cancer cells to establish at different organs sites are also expected to be different. In vitro comparison of primary PDA cell lines and matched distant metastasis revealed enhanced glucose entry into both glycolysis and the oxidative arm of the pentose phosphate pathway (PPP) in metastatic lines [98]. Several studies in other solid tumors such as melanoma [99], prostate [100] and breast [101] have shown that mitochondrial metabolism is linked to cancer metastasis. This dependency remains controversial, given that both mitochondrial dysfunction and the activation and inhibition of mitochondrial biogenesis have been shown to promote metastasis [99,100,101]. Differential dependency may be organ-site-specific, given that cancer cells that metastasize to the lung or lymph nodes have been reported to rely more heavily on mitochondrial ATP production [102,103,104], whereas those that metastasize to the liver seem to favor non-mitochondrial ATP production [104,105,106]. Taken together, these studies highlight the potential contribution of the TME to PDA metastatic progression, warranting further investigation. 

### 2.4. Dormancy

Metastatic dormancy can be defined as the time between the dissemination of cancer cells and the manifestation of a metastatic lesion. It is still unclear whether cancer cells leave the primary tumor in a dormant state, or if they disseminate in a pre-malignant state. Factors that govern dormancy and immune evasion remain elusive. A recent study suggested the involvement of endoplasmic reticulum (ER) stress in establishing dormancy in pancreatic cancer cells in vivo [107]. Specifically, these quiescent disseminated cancer cells (DCCs) exhibit unresolved ER stress along with a downregulation of the major histocompatibility complex class I (MHCI). In this setting, the use of small molecules to relief ER stress in combination with T cell depletion led to outgrowth of metastases in vivo [107]. This study has several implications in the development of therapies to prevent metastatic outgrowth in patients after removal of the primary tumor. In fact, data from this study suggest that post-operative hyperalimentation [108] (by balancing the levels of plasma cortisol after the surgery, and inducing an intact immune system), together with chemical chaperones, might play a role in clearing latent DCCs and suppressing the formation of metastasis after the removal of the primary tumor. 

## 3. Models of Pancreatic Metastatic Disease

As discussed above, the formation of metastasis is an extremely complex process that can be conceptually divided into three main phases: 1) intravasation of cancer cells from the primary tumor into circulation, 2) dissemination and survival of circulating tumor cells in the bloodstream, and 3) survival and colonization of disseminated cells in the distant site (Figure 1). In human PDA, the liver is the most common site of metastasis (accounting for over 60% of the patients), followed by lung and peritoneum metastasis (around 30%). Bone and adrenal secondary tumors account for approximately 10% of the metastasis in PDA patients [9,109,110]. 

In this section, we will provide an overview of the in vitro and in vivo models to study the metastatic process of PDA (summarized in Figure 1). 

### 3.1. In Vivo Models

#### 3.1.1. Murine Models

##### Genetically Engineered Mouse Models (GEMMs)

Genetically engineered mouse models (GEMMs) express tumor-driving genes in an immune-competent mouse. Thus, these models nicely recapitulate the histopathological features of PDA. However, unlike patients, PDA GEMMs die with, not from, metastatic disease [111]. 

• Transgenic Models

Transgenic models involve the ectopic expression of target genes in the host mouse genome. Tissue and/or cell type-specific promoters are often used to restrict the expression of the target gene spatially and temporally [112]. Several pancreatic cell-lineage-specific promoters have been used in GEMMs so far, including pancreatic and duodenal homeobox 1 (Pdx1) [112], elastase (Ela) [112], neurog3 (Ngn3) [113], and Ptf1 [111,112], among others. A thorough revision of the most common pancreas-specific Cre driver lines can be found in [112]. Ectopic expression of *Myc* under the elastase promoter drives liver metastasis in 20% of the mice [114], whereas the expression of the mouse polyoma virus middle T antigen in elastase-expressing cells in conjunction with the inactivation of tumor suppressor genes such as *p53*, *Smad4,* and *p16Ink4a* have been shown to stimulate the formation of highly metastatic pancreatic tumors [115]. Genetic ablation of the pigment epithelium derived factor (PEDF) in an *Ela-Kras^G12D^* mouse has also be shown to induce invasive pancreatic cancer [116].

While transgenic mice offer the advantages of being relatively fast to develop and breed, and allow the expression of human genes [117], the expression of the target gene occurs under foreign promoters at levels that do not necessarily represent the physiological expression level from its endogenous locus [117]. These limitations can be circumvented with the use of conditional knock-in or knock-out mouse models. 

• Conditional Gene Knock-in Models

Gene knock-in strategies offer the opportunity to express desired mutations in the gene of interest within its endogenous locus. Here, the engineered mutation lies downstream of a “Lox-STOP-Lox” (LSL) cassette and the interbreeding of mice carrying the mutant allele with a Cre driver mouse allows the expression of the target gene mutation in a tissue-specific manner. Mice expressing *Pdx1-Cre* and *LSL-Kras^G12D^* spontaneously develop metastatic adenocarcinomas at a low frequency [118]. 

The combination of oncogene activation with tumor suppressor inactivation has been a fruitful strategy in generating metastatic pancreatic disease models that closely resemble human PDA. The combination of activated *Kras^G12D^* expression with full body deletion of *p16Ink4a/p19Arf* has been reported to induce rapid progression of PanIN to invasive and metastatic PDA [119]. Similarly, conditional deletion of *p16Ink4a* in the context of *Pdx1-Cre*-driven *Kras^G12D^* expression drives the progression of pancreatic disease from PanIN to metastatic PDA [120]. Metastatic progression in this model is also accompanied by the loss of the *Kras* wild-type allele [120]. Conditional *Tgfbr2* deletion in the context of *Kras* activation (*Ptf1a^cre/+^;LSL-Kras^G12D/+^;Tgfbr2^flox/flox^*) leads to PDA with prominent desmoplasia [121]. Although most of the mice had to be sacrificed at an age in which no distant metastasis was observed, those that survived the longest demonstrated liver and lung metastasis, along with invasion to the diaphragm and the duodenum [121]. Although genetic loss of the *p53* tumor suppressor has been associated with metastasis in PDA, a direct comparison of mice bearing mutant *p53^R172H^* (*Pdx1-Cre;LSL-Kras^G12D^;p53^R172H/+^*) to conditional deletion of *p53* (*Pdx1-Cre;LSL-Kras^G12D^;p53^flox/flox^*) revealed that metastasis was observed only in *p53^R172H^* mutant-expressing PDA [122], suggesting that the *R172H* mutation is a *p53*-gain of function mutation that promotes PDA metastasis. Mice expressing *Kras^G12D^* in the context of double heterozygosity for *p53* and *p16Ink4a* or heterozygosity for *p19Arf* and *p16Ink4a* in the pancreas exhibited longer latency and higher propensity for metastasis relative to mice that express *Kras^G12D^* in the context of the homozygous deletion of *p53* or *p16Ink4a*/*p19Arf* separately, highlighting the cooperative role for double heterozygous *p16Ink4a* and *p19Arf-p53* in PDA progression [123]. Additional GEMMs to study PDA metastasis are summarized in Table 1. As presented in Table 1, although GEMMs can faithfully recapitulate the histopathological features of human primary PDA, the metastatic tropism of these models does not fully recapitulate the human disease (Table 1). 

##### Transplantation Models

Transplantation models consist of the implantation of human or mouse cells/tissues into recipient mice. Depending on where the cells are implanted, these models can be orthotopic (i.e., in the pancreas), or heterotopic (i.e., outside the pancreas). Cells engrafted through orthotopic transplantation can spread from the primary tumor to distant organ sites, therefore allowing the entire metastatic cascade to be modelled [111]. These transplant models provide the advantage of tractability and a relatively shorter and more predictable tumor latency [127]. In addition to orthotopic transplants, cancer cells can also be injected directly into circulation to model the steps of dissemination, extravasation, and colonization [111,127]. Heterotopic injections can be subcutaneous, intraperitoneal, intravenous, intra-splenic, or intra-cardiac. The site of colonization is dependent on the site of vascular injection [111]. For example, cells injected through the tail vein (intravenous) generally give rise to pulmonary metastasis, whereas intra-splenic injection generally gives rise to hepatic metastasis. For unbiased experimentation of tropism, intra-cardiac injection is favorable as it allows for the systemic dissemination to multiple sites [127].

Transplantation models can be syngeneic (allograft) or xenogeneic (xenograft). Allograft models allow the interrogation of metastatic dissemination in the context of an intact immune system, and therefore more faithfully recapitulate the TME. Tumor pieces or isolated cancer cells derived from GEMMs can be used to generate allograft models [128,129,130], which are characterized by a rapid and consistent development of tumors and up of 90% liver metastasis [129], thus making them more time- and cost-effective than GEMMs. The high frequency of metastasis in this model is likely a consequence of focal disease formation, which better resembles the sporadic mutations in KRAS found in human disease. 

Xenograft models involve the transplantation of human cancer cells or tumors into immune-compromised mice. Established cancer cell lines are a common source of material for transplant. However, since molecular and phenotypic properties may drift in culture, xenograft models using cancer cell lines do not always predict clinical responses [111]. Patient-derived xenografts (PDXs) represent a more favorable alternative, as they avoid in vitro selection pressures. In these models, patient tumor tissues are directly transplanted into immune-compromised mice for propagation in vivo [131]. Pancreatic PDXs have been shown to maintain the histology and metastatic potential of the patient-derived tumor [132]. These models can recapitulate the complexity of the TME in PDA, although the initial human stroma is gradually replaced with cells of the murine host [132,133]. A major drawback of the xenograft models is the requirement for a compromised adaptive immune system in order to prevent rejection by the host. This represents a major limitation when using these models to study metastasis, as the adaptive immune system is now known to play an important role in the selection of metastatic variants [134,135]. 

Care must be taken when selecting the appropriate model to use in a study. Subcutaneous xenograft mouse models do not constitute a good model to study PDA metastasis, as they have been shown to rarely metastasize [136], whereas orthotopically xenografted PDA frequently develop metastasis [137,138]. Indeed, in a 2015 study, Dai and colleagues compared two orthotopic xenograft mouse models with a subcutaneous tumor xenograft model and showed that the former develop metastasis in 80% of the mice, whereas the latter exhibits no metastasis [138]. Moreover, different commercially available pancreatic cancer cell lines exhibit varying degrees of metastatic activity, ranging from 0 to 90% [137]. The data from these studies are summarized in Table 2. 

##### Models to Study PDA Early Dissemination

Classically, tumor dissemination is viewed as a late event in the disease progression, after the formation of a primary tumor. However, emerging data support the idea that cancer cells can spread to distant sites even before the establishment of a primary tumor [30,31]. Indeed, using the *Kras^G12D^;p53^fl/+^; Pdx1-Cre;Rosa^YFP^* (KPCY) mouse model, it has been shown that YFP-positive cells can be found in the circulation and liver parenchyma of KPCY PanIN-bearing mice in the absence of a frank tumor. These cells have undergone EMT and exhibit a mesenchymal phenotype, showing increased survival and self-renewal in vitro [139]. These findings reveal that EMT precedes tumor formation and they were further validated in a small human cohort of patients, identifying circulating pancreatic cancer cells in 33% of patients with cystic precancerous lesions and 73% of patients with PDA [140]. The molecular characterization of pancreatic cancer cells in circulation will be central to understanding the properties of early disseminated PDA. 

Circulating tumor cells (CTCs) are rare cells shed by solid tumors into the systemic circulation at an estimated frequency of 1:500,000–1:1,000,000 circulating cells, with a half-life of between 1 and 2.5 h [141]. The utility of CTCs to predict metastatic disease in human PDA disease is an area of active research [142]. Currently, CellSearch is the only FDA-approved platform for CTC detection and is based on EpCAM expression, which is expressed in both normal and malignant epithelial cells [142]. In the absence of molecular markers to distinguish malignant CTCs from circulating normal epithelial cells, the molecular characterization of CTCs has remained challenging. Wnt2 has been shown to be enriched in metastatic PDA patients and has been proposed to be a potential marker of pancreatic CTCs [143]. Additional markers of this nature that are expressed in early stage disease will be crucial for the detection and hence molecular characterization of early disseminated PDA. Advances in the sensitivity of CTC capture, along with single cell-based analysis, will allow the interrogation of the factors that mediate the early intravasation of pancreatic cancer cells into circulation, and their survival in distant organ sites. Several reviews on the advances in the capturing and identification of circulating tumor cells in general are available [144,145,146,147]. 

#### 3.1.2. Zebrafish

The zebrafish (*Danio rerio*) provides many advantages in cancer research over in vivo murine models due to its relatively low maintenance cost, work feasibility, and tractability. Many pathways of tumor progression are shared between mammals and the zebrafish [148]. The optical transparency of the *casper* zebrafish line [149] makes it possible to visualize tumor progression and metastasis using microscopy. Lastly, due to an under-developed immune system in the zebrafish larvae, most transplanted cancer cells can survive and form metastasis in this system [150]. 

Several studies have shown the potential of using the zebrafish to study pancreatic cancer metastasis [151,152,153]. In 2008, Park and colleagues used the zebrafish model to study the effects of *KRAS* activation in pancreatic progenitor cells [153]. This study showed that *KRAS* activation leads to the formation of invasive pancreatic cancer with a similar aggressive behavior as human pancreatic cancer, including the propensity to metastasize. Using this model system, Weiss et al. showed that retinoic acid receptor antagonists repress microRNA-10a, blocking the metastatic potential of pancreatic cancer cells, and establishing a role for microRNA-10a in pancreatic metastasis formation [151]. These results were further validated in a later study showing that microRNA-10a is overexpressed in a subset of pancreatic patients, and that it promotes the invasiveness of the cancer cells [154]. 

#### 3.1.3. Chick Embryo

The chick embryo is a simple alternative to the more complex and expensive mouse models. Because of the thin, accessible chorioallantoic membrane (CAM), this system allows for the easy imaging and analysis of migration and metastasis in vivo. Moreover, imaging does not require surgery or anesthesia, as with rodent models [155,156]. Additionally, this is a naturally immune-deficient system, which allows transplantation of tumor cells of different tissue and species origin [157]. Using this model, Fujimura and colleagues reported that the translation initiation factor 5A (eIF5A) is necessary for PDA metastasis, as knocking down its expression reduces the number of metastasis in the liver [158]. In a different study, inoculation of PSCs together with PANC-1 cancer cells promoted invasion of the CAM and tumor formation, thus supporting the concept that PSCs promote the progression of PDA metastasis [159].

### 3.2. In Vitro Systems

In vitro models of the different phases of the metastatic process in pancreatic cancer have been used as cost- and time-effective alternatives to animal models. Importantly, these models allow for the in-depth molecular interrogation of the effect of chemical, physical and mechanical parameters on cell migration and invasion. 

#### 3.2.1. Two-Dimensional Monolayer Culture

Several methods have been developed to study the migration and invasion of 2D (monolayer) cancer cells. One such method is the scratch healing assay, in which a central scratch is created across a confluent monolayer of cells, and the measurement of cell migration into the wound is performed through microscopy [160]. This method is not suited for suspension cells or for the analysis of chemotaxis, but provides a fast and inexpensive approach to measuring migration kinetics in real time, and to assess the interaction between tumor cells and different extracellular matrix substrates [160]. As an alternative, cell migration can also be studied through cell exclusion assays, in which cancer cells are seeded into inserts that are removed once a confluent monolayer is formed. This avoids potential cell damage created when making the scratch and increases reproducibility [160]. 

Transwell and Boyden chamber assays are widely used methods to simulate migration and invasion of cancer cells across the epithelium [160]. In these methods, cancer cells are placed in an insert composed of two chambers separated by a porous membrane, and their capacity to transmigrate from one chamber to the other is evaluated [160]. To mimic tumor invasion, a layer of ECM (such as matrigel) is included so that invasive cells must degrade the matrix to migrate [160,161]. Despite providing several advantages over in vivo methods (including the capacity to fine-tune experimental parameters, and being relatively inexpensive and easy to use), these methods represent endpoint studies, are limited in their capacity to study multicellular interactions, and do not provide information beyond the number of migrated cells [160,162]. Optical mobility assay devices such as the TaxiScan, on the other hand, can be used to obtain additional information on migrating cells, including morphology, directionality and velocity [163]. Using these methods, several studies have shown that only a small fraction of pancreatic cancer cells are capable of invading the ECM [164,165,166,167,168,169]. These cells upregulate the expression of protein tyrosine kinase 6 (PTK6) [164], nitric oxide (NO) levels [165], and the activation of the RhoA and PI3K-AKT pathways [166]. 

To probe the complex interactions between cancer cells and cells of the TME in the process of metastasis, the assays above can also be done in the context of direct or indirect co-cultures. Typical co-culture experiments involve seeding of not only cancer cells, but also of stromal cells. In this scenario, cancer cells can receive the physical, mechanical and biological signals from the surrounding environment (such as cytokines and growth factors) [170,171]. Upon co-culture with patient-derived PSCs, pancreatic cancer cells exhibit an increase in EMT markers and migration, further highlighting a role for PSCs in pancreatic cancer metastatic progression [172]. 

To better mimic physiological conditions in vivo, microfluidic assays can be used to explore the formation of metastasis in a more physiologically relevant manner. Recent advances in the microfluidics field have allowed the investigation of three important aspects of cell migration and metastasis development: flow/shear stress, chemical gradients, and the complex interaction between multiple cell types [162,173]. In conjunction with microscopy-based time lapse imaging, this system is a powerful tool to investigate the biophysical parameters that drive PDA metastasis [160,162]. 

#### 3.2.2. Three-Dimensional Organoid Cultures 

Despite being time- and cost-effective, cells grown in monolayer lack the structural complexity and architecture of human tissues. Three-dimensional organoid cultures provide an alternative, given their ability to maintain cell polarity and interaction with an extracellular matrix [174]. Currently, different approaches to culturing pancreatic organoids from normal and tumor tissue have been developed [175,176,177,178] and are now an invaluable resource for fundamental and applied studies of pancreatic cancer, with great potential in drug screening, and tumor-host interaction. 

As patient-derived organoids closely resemble the molecular features of the original tumor and maintain intra-tumor heterogeneity [176,179], this also provides a unique opportunity to perform deep molecular pairwise comparisons between murine or patient-derived primary tumors and distant metastases ex vivo. Moreover, patient-derived organoids represent an attractive tool to study the progression of pancreatic cancer in vivo. In fact, xenograft models involving the transplantation of pancreatic tumor organoids have been shown to generate the full spectrum of pancreatic cancer progression, from the initial PanIN stages, to invasive adenocarcinoma, followed by metastasis [177]. 

## 4. Future Outlook

Pancreatic cancer is a disease characterized by an early and rapid metastatic process. The early dissemination of cancer cells can be partially explained by the localization of the pancreas close to the spleen and kidney, as well as large blood vessels [142]. However, we currently lack deep molecular insight into the metastatic process of pancreatic cancer. 

Pancreatic cancer usually metastasizes to the liver, lungs, and peritoneum. However, very few studies have focused on trying to understand the mechanisms behind PDA metastatic organotropism (i.e., the development of metastasis in particular organs or tissues). This is imperative given that the location of the metastases affects the clinical outcome for the patient as, for example, patients with lung metastases have an improved outcome compared to those with liver metastases [109]. A previous study by Hoshino et al. suggested that tumor-secreted exosomes are sufficient to direct cancer cells to specific organs, due to exosomal integrin fusion with target cells in a specific organ [180]. Organoid culture and its culture supernatant could represent a potentially useful way to further characterize the mechanism driving this process as well as the differential cargo of disease stage-specific or organ site-specific exosomes, and their role in tumorigenesis. Recent work by Reichert and colleagues using GEMMs to investigate the regulation of metastatic organotropism in PDA showed that the formation of liver and lung metastasis in PDA is dependent on p120catenin (p120ctn) [181]. In fact, the authors demonstrated that biallelic p120ctn loss is necessary for lung metastasis and prevents liver metastasis, whereas monoallelic p120ctn loss accelerates the formation of metastasis in the liver. Overall, the scarcity of data on pancreatic cancer organotropism highlights the need to better understand this process. The ability to correctly predict the organ site of future metastasis and metastatic predisposition in pancreatic patients will allow us to cater specific therapeutic strategies to different patient groups. Model systems to interrogate the process of organotropism in vivo would be central towards this goal. 

Much of our current understanding of PDA metastasis involves vascular migration. Lymphatic migration, on the other hand, is very poorly studied. This is pertinent given that patients with lymph node metastasis have worse survival rates than those without it [182]. As lymph node dissemination has been characterized as an early event in tumor development [183], one could hypothesize that the lymph node might act as a reservoir for further seeding into other organs. Moreover, what makes the lymph node an ideal place for dissemination and whether the cancer cells play a role in preparing its microenvironment there are crucial questions to which we still need to find the answers to. The development of models to interrogate lymphatic migration, such as that described by Xiong and colleagues [184], is a step forward towards that goal. 

A major obstacle underlying the clinical challenges in pancreatic cancer is our limited understanding of the molecular mechanisms of PDA metastasis. This has been partially attributed to the lack of proper models to study the metastatic progression of this disease. Technological advances in this area will be central to the development of novel therapeutics that target PDA metastatic dissemination.

## Figures and Tables

**Figure 1 genes-11-00006-f001:**
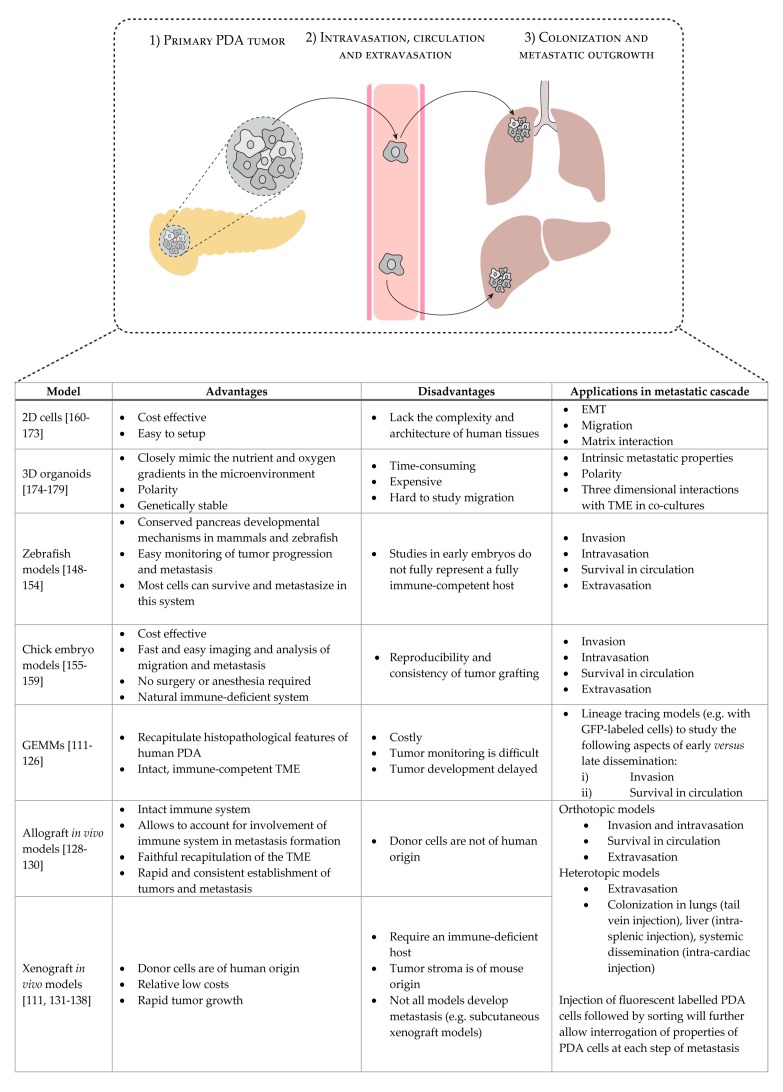
Overview of the current models to study metastatic PDA, and the major advantages and disadvantages of each model. Pancreatic ductal adenocarcinoma (PDA); Epithelial-to-mesenchymal transition (EMT); Tumor microenvironment (TME).

**Table 1 genes-11-00006-t001:** Metastatic rate of the different GEMM models.

Model	Promoter	Tumor Latency	% Metastasis	Reference
*Ela-myc*	Elastase	2–7 months	Peritoneal 68% Liver 20%	[114]
*Pdx1-Cre;LSL-Kras^G12D^; Ink4a/Arf^lox/lox^*	Pdx1-Cre	5 weeks	Renal lymph node 4.2% Liver 12.5% Peripancreatic Lymph node 4.2%	[119]
*p16^−/−^;LSL-Kras^G12D^;Pdx1-Cre*	Pdx1-Cre	6–24 weeks	Liver 31.8% Lymph node 13.6% Lungs 4.5%	[120]
*Pdx1-Cre;LSL-Kras^G12D^;p53^R172H/+^*	Pdx1-Cre	10 weeks	Liver 65%	[122]
*LSL-Kras^G12D^;LSL-p53^R172H/+^;Pdx1-Cre*	Pdx1-Cre	N/A	Liver 63% Lungs 44% Diaphragm 37% Adrenal 22%	[124]
*Ptf1a^cre/+^;LSL-Kras^G12D/+^;Tgfbr2^lox/lox^*	Ptf1a-Cre	N/A	Liver 12% Lung 8%	[121]
*Pdx1-Cre; LSL-Kras^G12D^;p16/p19^lox/lox^*	Pdx1-Cre	8.5 weeks	11%	[123]
*Pdx1-Cre; LSL-Kras^G12D^;p16/p19^lox/+^*	Pdx1-Cre	34.2 weeks	69%	[123]
*Pdx1-Cre; LSL-Kras^G12D^;p53^lox/lox^;p16^+/+^*	Pdx1-Cre	6.2 weeks	0%	[123]
*Pdx1-Cre; LSL-Kras^G12D^;p53^lox/lox^;p16^+/–^*	Pdx1-Cre	6.5 weeks	0%	[123]
*Pdx1-Cre; LSL-Kras^G12D^;p53^lox/lox^;p16^–/–^*	Pdx1-Cre	7.2 weeks	20%	[123]
*Pdx1-Cre; LSL-Kras^G12D^;p53^lox/+^;p16^+/+^*	Pdx1-Cre	21.8 weeks	33%	[123]
*Pdx1-Cre; LSL-Kras^G12D^;p53^lox/+^;p16^+/–^*	Pdx1-Cre	14.7 weeks	25%	[123]
*Pdx1-Cre; LSL-Kras^G12D^;p53^lox/+^;p16^–/–^*	Pdx1-Cre	13.1 weeks	25%	[123]
*Pdx1-Cre; LSL-Kras^G12D^;p53^+/+^;p16^–/–^*	Pdx1-Cre	18.3 weeks	33%	[123]
*Pdx1-Cre; LSL-Kras^G12D^*	Pdx1-Cre	57 weeks	67%	[123]
*Ptf1a(P48)-Cre; Kras^G12D/+^; MUC1.Tg*	Ptf1a(P48)-Cre	26 weeks	60% (lung and liver metastasis)	[125]
*Pdx1-Cre; Kras^G12D/+^; Rb^loxP/loxP^*	Pdx1-Cre	2 weeks–5 months	0%	[126]

For each GEMM model, the promoter, tumor latency and percentage of metastasis is indicated. The percentage of metastasis is in reference to the entire *n* included in the study. N/A: not available.

**Table 2 genes-11-00006-t002:** Degree of metastatic activity of different cell lines in transplantation models.

Cell Line	Model	Liver Metastasis	Lungs Metastasis	Lymph Nodes Metastasis	Reference
Capan-1	Orthotopic	86%	29%	43%	[137]
Capan-2	Orthotopic	56%	0%	0%	[137]
HPAF-II	Orthotopic	13%	0%	13%	[137]
CFPAC	Orthotopic	50%	20%	40%	[137]
HPAC	Orthotopic	14%	14%	14%	[137]
Panc-1	Orthotopic	88%	0%	50%	[137]
AsPC-1	Orthotopic	80%	80%	90%	[137]
AsPC-1	Orthotopic	20%	N/A	N/A	[138]
AsPC-1	Subcutaneous	0%	N/A	N/A	[138]
MPanc96	Orthotopic	89%	56%	67%	[137]
BxPC-3	Orthotopic	67%	0%	17%	[137]
Hs766T	Orthotopic	40%	20%	10%	[137]

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
