# Peer review of "Metastasis in Pancreatic Ductal Adenocarcinoma: Current Standing and Methodologies"

_genes, 2019, doi:10.3390/genes11010006_

Round 1

Reviewer 1 Report

In the present paper, Marina Ayres Pereira and Iok In Christine Chio are reviewing the litterature regarding pancreatic cancer studies with a strong focus on metastasis.

While metastasis is a major problem in pancreatic cancer, as highlighted in the introduction, it always seems like the scientific community in this field was more focused on describing mechanisms related to the primary tumor. I think this review may contribute to draw attention on that.

The introduction is very well balanced and not too long which allow to go straight to the science. Then, the known mechanisms contributing to metastasis in PDA are well described. There is also a good and complete description of the different models and technical possibilities for studying PDA metastasis.

This is overall a very good review that has the advantage of going straight where it should be and is thus very light and pleasant to read (contrary to most of what we usually see in the litterature).

I would like to suggest only one minor modification: In the paragraph regarding xenograft models the limitations are well stated concerning the source of cancer cells and the injection site. However I think it is important to add one or two sentences highlighting the lack of a fonctionnal adaptive immune response in those models. Indeed the selection of cancer cell variants by the T cell response both in the primary tumor and metastatic site seems to play a key role thus relativising the importance of mechanisms described in xenograft models.

Author Response

Reviewer 1: In the present paper, Marina Ayres Pereira and Iok In Christine Chio are reviewing the literature regarding pancreatic cancer studies with a strong focus on metastasis.
While metastasis is a major problem in pancreatic cancer, as highlighted in the introduction, it always seems like the scientific community in this field was more focused on describing mechanisms related to the primary tumor. I think this review may contribute to draw attention on that.

The introduction is very well balanced and not too long which allow to go straight to the science. Then, the known mechanisms contributing to metastasis in PDA are well described. There is also a good and complete description of the different models and technical possibilities for studying PDA metastasis.
This is overall a very good review that has the advantage of going straight where it should be and is thus very light and pleasant to read (contrary to most of what we usually see in the literature).

I would like to suggest only one minor modification: In the paragraph regarding xenograft models the limitations are well stated concerning the source of cancer cells and the injection site. However, I think it is important to add one or two sentences highlighting the lack of a functional adaptive immune response in those models. Indeed, the selection of cancer cell variants by the T cell response both in the primary tumor and metastatic site seems to play a key role thus relativizing the importance of mechanisms described in xenograft models.

Response: We appreciate the reviewer’s positive feedback, and would like to thank the reviewer for raising this comment. We have revised the manuscript to address the concern raised (lines 314- 318 with all markup; lines 309-313 with no markup).

Reviewer 2 Report

The review titled “Metastasis in Pancreatic Cancer: advances in the field and methodologies” by M. A. Pereira and I. I. C. Chio discusses the problem of metastasis in pancreatic ductal adenocarcinoma and different methods which are utilized to study this phenomenon. The review is well written and flows logically, however, there are several issues, and the paper needs some major revision. Overall, the manuscript is within the scope of the journal.

General Comments

Pancreatic ductal adenocarcinoma (PDA) is defined twice, in line number 26 and 29. Do neuroendocrine tumors not metastasize, and that is why not the part of this paper? Please include a rationale in the paper for not including neuroendocrine tumors.

Specific Comments:

The review does not accurately describe the advances in the field of pancreatic cancer metastasis. Please include a separate section on advances in the field of pancreatic cancer metastasis. Otherwise, please remove the word “advances” from the title.

Though, the section on “factors governing metastasis” cites example from pancreatic cancer studies, but does not distinguish the specific problems in pancreatic cancer metastasis from any other cancer metastasis as such. This section should be modified to be more specific on pancreatic cancer metastasis.

Please include the section on models supporting evidence of early and late metastasis.

Again, the section on “models of pancreatic metastatic disease” including overview information in Figure 1 is, in general, accurate for any cancer metastasis and less specific to pancreatic cancer metastasis. Please add information in this section describing advantages and disadvantages of these models in light of different stages of pancreatic cancer metastasis. For example models that are suitable to study the steps of intravasation/extravasation or survival in circulation. It will add specificity to the review and will make it more interesting to read.

Author Response

Reviewer 2
The review titled “Metastasis in Pancreatic Cancer: advances in the field and methodologies” by M. A. Pereira and I. I. C. Chio discusses the problem of metastasis in pancreatic ductal adenocarcinoma and different methods which are utilized to study this phenomenon. The review is well written and flows logically, however, there are several issues, and the paper needs some major revision. Overall, the manuscript is within the scope of the journal.

General Comments

ï‚· Pancreatic ductal adenocarcinoma (PDA) is defined twice, in line number 26 and 29. Do neuroendocrine tumors not metastasize, and that is why not the part of this paper? Please include a rationale in the paper for not including neuroendocrine tumors.

o Response: We have removed the second definition of PDA (line 34 with all markup, and line 33 with no markup). We thank the reviewer for their suggestion regarding neuroendocrine tumors, and have revised the manuscript to provide a better rationale for focusing on PDA in the current manuscript (lines 28-33 with all markup, lines 26-32 with no markup). Briefly, as mentioned in the first paragraph of the review (lines 25-28 with all markup, lines 24-27 with no markup), only a small percentage of patients (7%) are diagnosed with pancreatic neuroendocrine tumors (PNET) (Antonello D. et al., 2009 and Amin S. et al., 2006). In addition to the low frequency, most neuroendocrine tumors are characterized as indolent (Ro C, et al., 2013). Despite the fact that these tumors can metastasize (most commonly to the liver), treatment options are available to patients with metastatic PNET, including liver transplant, chemotherapy and locoregional therapy (Ro C, et al., 2013). This is in contrast to PDA patients with liver metastasis, for whom very limited treatment options are available. Taken together, we have selected to focus on PDA in this review, but have also directed interested readers to a relevant review on neuroendocrine tumors of the pancreas.

Specific Comments:
ï‚· The review does not accurately describe the advances in the field of pancreatic cancer metastasis.

Please include a separate section on advances in the field of pancreatic cancer metastasis. Otherwise, please remove the word “advances” from the title.

o Response: We thank the reviewer for this comment, and revised the title of the manuscript as suggested. We have changed the title from “Metastasis in Pancreatic Cancer: advances in the field and methodologies” to “Metastasis in Pancreatic Ductal Adenocarcinoma: current standing and methodologies” (lines 2-4 with all markup, lines 2-3 with no markup).

ï‚· Though, the section on “factors governing metastasis” cites example from pancreatic cancer studies, but does not distinguish the specific problems in pancreatic cancer metastasis from any other cancer metastasis as such. This section should be modified to be more specific on pancreatic cancer metastasis.

o Response: We thank the reviewer for this comment. We agree with the reviewer and have revised this section accordingly. In addition to describing the various factors governing metastatic dissemination that are shared among different cancer types, we have revised this section to further highlight features that are reported for pancreatic tumors. This includes the role of genomic instability, cancer stem cells, dormancy, metabolism, clonality, and times of dissemination that are unique to PDA metastasis (lines 74-211 with all markup, and lines 73-206 with no markup).

ï‚· Please include the section on models supporting evidence of early and late metastasis.
o Response: We thank the reviewer for this suggestion, and have added a subchapter entitled “Models to study PDA early dissemination” (lines 330-355 with all markup, and lines 325-350 with no markup) where we briefly describe the contribution of lineage tracing models and

circulating tumor cells (CTCs) to the study of early tumor dissemination.

ï‚· Again, the section on “models of pancreatic metastatic disease” including overview information in Figure 1 is, in general, accurate for any cancer metastasis and less specific to pancreatic cancer metastasis. Please add information in this section describing advantages and disadvantages of these models in light of different stages of pancreatic cancer metastasis. For example, models that are suitable to study the steps of intravasation/extravasation or survival in circulation. It will add specificity to the review and will make it more interesting to read.

o Response: We have included a new column in Figure 1 entitled “Applications in metastatic cascade” in which we identify for each model the step of PDA metastatic dissemination it is applicable for. This way, we believe Figure 1 provides a general overview of the advantages and drawbacks of the different available models, as well as their specific applications within the metastatic cascade.

Round 2

Reviewer 2 Report

The authors have addressed all the concerns raised and can be accepted in the current form